# Diagnosing and Balancing Approaches of Bowed Rotating Systems: A Review

Nima Rezazadeh, Alessandro De Luca *, Giuseppe Lamanna and Francesco Caputo

Department of Engineering, University of Campania "Luigi Vanvitelli", 81031 Aversa, Italy
* Correspondence: alessandro.deluca@unicampania.it

**Abstract:** Driven/driving shafts are the most important portion of rotating devices. Misdiagnosis or late diagnosis of these components could result in severe vibrations, defects in other parts (particularly bearings), and ultimately catastrophic failures. A shaft bow is a common problem in heavy rotating systems equipped with such attachments as blades, discs, etc. Many factors can cause the shaft bending; this malfunction can be temporary, such as the bow resulting from a rotor gravitational sag, or can be permanent, such as shrink fitting. Since bending effects are similar to those induced by the classic eccentricity of the mass from the geometric center, i.e., unbalancing, distinguishing the differences in dynamic behaviors, as well as the symptoms, can be a labor-intensive and specialized task. This article represents a review of almost all the investigations and studies that have been carried out on the diagnosing and balancing of bowed rotating systems. The articles are categorized into two major classes, diagnosing and balancing/correcting approaches to bowed rotors. The former is divided into three subclasses, i.e., time-domain, frequency-domain, and time–frequency-domain analyses; the latter is divided into three other sub-sections that concern influence coefficient, modal balancing, and optimization method in correcting. Since the number of investigations in the time domain is relatively high, this category is subdivided into two groups: manual and smart inspection. Finally, a summary is provided, as well as some new research prospects.

**Keywords:** rotating devices; defects; shaft bow; diagnosing; balancing

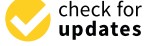



## 1. Introduction

A drastic level of vibration produces noise and reduces machine performance; it diminishes the machine's service life and reliability and can even result in dangerous difficulties. As a result, many sophisticated companies tend to have optimized maintenance schedules [1]. Assembly fault, asymmetric geometry, material inhomogeneity, and manufacturing tolerance are all possible causes of a defect [2].

For a variety of reasons, the shaft that serves as the primary transmission component in various rotor systems is not always as straight as it should be. Sometimes the shaft is assembled from a line to a curve in high-load-carrying rotor-bearing systems, such as marine propulsion shafting, to reduce the load on bearings that are close to the heavy propeller, so the shaft is subjected to additional bend forces that run parallel to the section [3]. On the other hand, unwanted bends in shafts can occur owing to a variety of factors, including creep, thermal distortion, or a strong imbalance force. Excessive heat, length, or physical bend can all cause the shaft bending. Figure 1 shows a schematic of a simple rotor-bearing-disk system that suffers a bowed shaft and its cross-sectional view. An unexpected failure of spinning machines due to a bent shaft might result in considerable financial losses. Depending on the extent and location of the bend, a bent shaft causes excessive vibration in a machine [3,4]. In industry, approximately 35% of breakdowns of rotating devices have resulted from bowed shafts and misalignment [5,6]. As a result, a proper shaft bow diagnosis, as well as a balancing/correction procedure, appear mandatory.

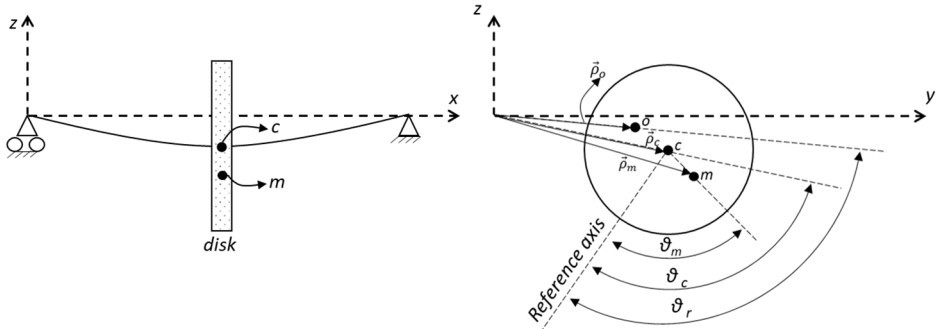

**Figure 1.** Schematic of a simple rotor-bearing-disc system and its cross-sectional view.

This paper proposes, for the first time, an extensive review of the state of the art on bowed rotor systems, and it can be considered as a guide for researchers who are interested to expand modern procedures in distinguishing and balancing bowed rotary devices.

In the right aforementioned figure, $\vartheta_r$, $\vartheta_m$, and $\vartheta_c$ are the phase angles of the residual bow vector, disk unbalancing vector, and the shaft center line vector with respect to the reference axis, respectively. $m$, $c$, and $o$ are the indicators of the mass center of the disk, center of the shaft, center of the bow, respectively. Furthermore, the shaft elastic deflection, residual bow, and unbalancing deflection vectors are represented by $\vec{\rho}_c$, $\vec{\rho}_o$, and $\vec{\rho}_m$, subsequently.

## 2. Materials and Methods

The paper is organized into two main categories: fault diagnosing and balancing/correction procedures, as clearly shown in Figure 2.

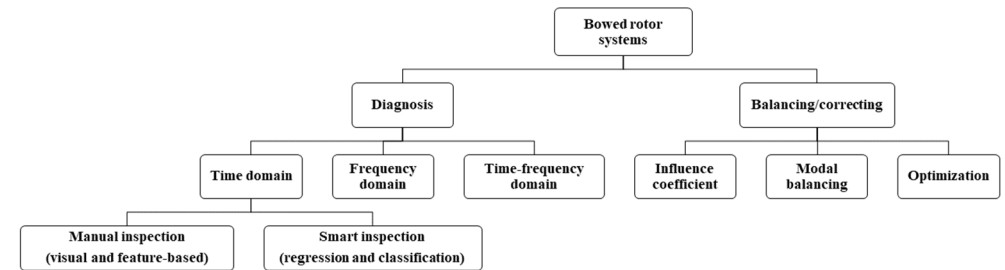

**Figure 2.** The employed categorization in this article.

### 2.1. Diagnosing

The process of fault diagnosis consists mainly of three phases: detection, isolation, and identification, also denoted FDII [7]. Detection means the capability to identify system anomalies; isolation concerns the capability in determining the type and location (which component) of a malfunction; identification focuses on the defect's severity [8].

Fault diagnosis in rotary machines can be performed thanks to a wide spectrum of traditional and modern methods, ranging from listening to the sound of the working device to using deep neural networks to automatically classify defects [9–11]. Using examples of unbalanced, cracked, and bent rotor systems, Edwards et al. [12], in 1998, reviewed fault diagnosing works that had been carried out in those situations; in [13], Walker, Perinpanayagam, and Jennions gave a brief review of the recent advances in the field of fault diagnosis methods for rotating machines. The work carried out in the field of bowed rotor diagnosis using vibration-based methods can be divided into three major categories: time domain, frequency domain, and time–frequency domain. The three aforementioned categories are used to group the present investigations that have been carried out to diagnose bowed rotor systems. Additionally, since the number of previous works in the time-domain is much higher than the two other categories, this domain is subdivided into manual and smart inspections.

### 2.1.1. Time Domain

Signals captured by transducers, e.g., displacement, velocity, and acceleration sensors from a system in industrial applications, are mostly in the time domain, and time-domain analysis is the study of these raw signals without any transformation into the frequency domain. Although in the current computerized era there is more tendency to use automatic fault diagnosing, time-domain analysis can be performed both in manual and smart manners [14].

Two different operating conditions, "steady-state" and "transient", can be considered when vibration-based methods are used to assess the health condition of a damaged rotor system. The transient signal is only measurable when the rotating acceleration is changing, i.e., when the system is starting up or shutting down. In contrast, the former signal can be measured throughout the system's operation after it starts up and before it shuts down [15]. A bowed rotor system's vibration signals, including transient and steady-state segments, are shown in Figure 3. The related areas for transient (startup), steady-state operation prior to the first critical speed, during the first critical speed, and steady-state operation following the first critical speed are regions 1, 2, 3, and 4.

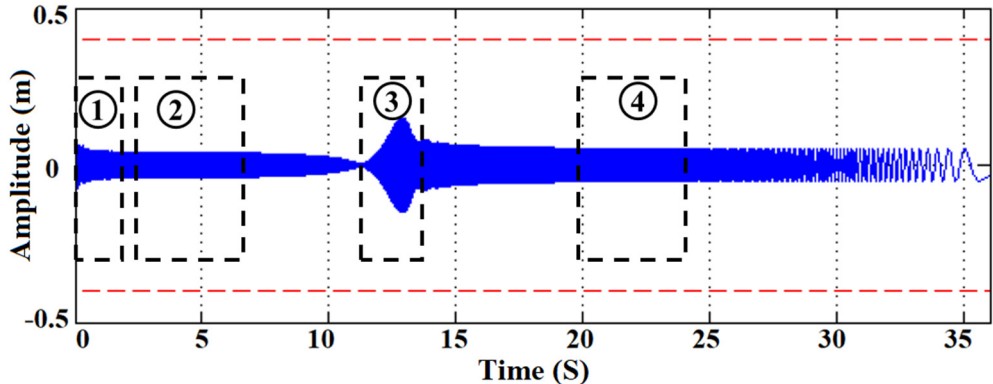

**Figure 3.** Different segments of the transient and steady-state signal of a bowed rotor system [16].

Manual Inspection

Manual time-domain analysis can be divided into two subfields based on visual and feature-based inspections. Visual inspection means comparing raw vibration signals of a faulty system with a healthy one in the same frequency. Because real signals are usually low-amplitude and polluted with background noise, visual inspection is not considered a reliable procedure to distinguish differences (damages), but this procedure is low-cost and not complicated. Furthermore, this approach can be effective if a specific segment of the response signal is aimed to be examined, for instance, around the system's critical speeds; in the case of real-world signals, firstly, a suitable noise cancellation method should be performed [17].

On the other hand, feature-based inspection in time domain can be employed both in manual and automatic manners. To extract features in time-domain analysis, there are two main approaches, e.g., statistical functions and advanced techniques. As illustrations of the former, peak amplitude, root mean square amplitude, and variance can be mentioned. Time-synchronous averaging and filter-based methods are examples of the latter [17].

Throughout the history of diagnosing bowed rotors, many researchers have used time-domain analysis both in terms of visual and feature-based inspection.

Probable effects of different amounts of the residual bow, phase angle, rotational speeds, and system damping ratio on the behavior of the bowed rotor systems have been evaluated by a number of researchers. Although vibrations caused by pure unbalance and a bowed shaft can both produce synchronous excitation, there are some differences between the two situations, particularly in the phase angle during resonance.

Nicholas, Gunter, and Allaire [6] compared the peak responses of unbalanced and bowed rotors for various amounts of bow amplitude, phase angle, and damping ratio. It was demonstrated that the vibration amplitude can be zero if two conditions are satisfied: first, the residual bow and unbalance must be perfectly out of phase (the phase difference must be 180°); and second, the rotation velocity should be equal to the square root of the ratio of the residual bow amplitude to imbalance eccentricity. A technique for calculating the ratio between the residual bow and unbalance eccentricity was also presented. The residual bow is smaller than the unbalance eccentricity when the response amplitude tends to be minimum below the critical speed. In contrast, it may indicate that the residual bow is larger than the unbalance eccentricity if the response amplitude decreases to a minimal level in the region above the critical speed.

Influences of three faults, i.e., unbalancing, initial bow, and disc skew, were investigated by Shiau and Lee [4] for a simply supported rotor system. Furthermore, the possible impacts of changing the disc position alongside the shaft were investigated. In the case of a thick disc, the square root of the ratio of initial bend magnitude to lumped imbalance eccentricity was found to be equal to the first self-balancing speed, and in the case of a thin disc, it was also shown to be equal to the self-balancing speed. The imparted moments acted as restoring moments in the thin disc case, pitching the disc in the opposite direction of the initial skew. The applied moments, on the other hand, tended to raise the disc pitch in the same direction as the original skew in the thick disc casing. The first critical speed shifted somewhat as the moment of inertia difference decreased, but the second critical speed varied dramatically as the moment of inertia difference decreased.

Moreover, the critical speed was discovered to be only a function of the disc location between bearings, also the moment of inertia differential. The magnitude of the vibration was observed to reach its maximum level when both the disc and the maximum initial bow of the shaft were placed in the center of the shaft. Both the translational and rotational vibration magnitudes decreased as the spinning speed raised when the damping was further than 30%.

In [18], Ehrich stated that the speed at which the rotating system operates can change the effects of the shaft bow. If the rotor system works below the critical speed, the effects of the unbalancing are smaller than the shaft bow. The effects of these two defects, i.e., shaft bow and unbalance, are the same at the critical speed, although at speeds more than the critical one, the influence of the shaft bow is masked by the effects of unbalance.

Edwards, Lees, and Friswell [19] proposed a novel process to identify parameters of the excitation forces and the foundations of a bowed rotating system in a single rundown or runup. Moreover, the shaft's bent profile was estimated with reasonable accuracy. Using this procedure, the magnitude of unwanted vibrations resulting from unbalancing and bowed shaft neutralized by about 92%. It was stated that the vibrations in the bent shaft have a 180° phase angle difference at resonance speed.

In [20], Rao studied the influences of various conditions of an initial bend on the behavior of rotor systems. It was stated that at resonance, the response phase shift is independent of the damping ratio, but changing the amount of the residual bow can alter the phase angle. The whirl amplitude decreases in the slow runout operation, parallel to a rise in the speed, if the bow and unbalancing are in the opposite location and the amount of bow is smaller than the eccentricity. When the bow amplitude exceeds the eccentricity, two different conditions can occur. If the bow amplitude is at most twice the eccentricity, the rotor whirl amplitude falls to zero after resonance and then rises to a unit value as the speed increases. Because the amplitudes have already fallen to very low values, it may be difficult to notice the trend in the previous note for large values of rotor bow (further than twice) in comparison to eccentricity. There were differences in the observed trend between low-speed and high-speed operations. Because the unbalance force is proportional to the rotor system's angular velocity, the bowed force was dominant at low speeds; however, the unbalance force could mask the bowed force effects at high speeds. The phase angle in the slow shutting down (transient signal) presents the angle of the bow position to the

imbalance eccentricity. If the phase angle is zero or close to it, the rotor may not have a bow, or it may have a bow in the same direction as the unbalance. The phase angle for rotors with a bow magnitude less than the eccentricity diminishes with increasing speed from the slow runout condition up to roughly the self-balancing speed, then increases. In [21], Rao and Sharma experimentally confirmed the aforementioned results, too.

Lin and Lei [3] developed a mathematical model of a bowed rotating system to investigate the effects of a bowed component (shaft) on the vibrations of the bowed and straight parts of the system. The system was assumed to have several different initial bends, also various operation speeds, then vibration signals were analyzed in the disc position and in the shaft in the time domain. The vibration of the bent part was primarily affected by the bending force, while the vibration of other straight parts was less affected. The intensity of the vibration response was mostly governed by the amount of bend. The vibration mode was generally stationary, with a lateral and vertical difference.

If the cooling process in a rotor system is not performed correctly, e.g., the system does not work in turning gears, then restarts immediately, the occurrence of a thermal bow is plausible. In this case, the vibration amplitude can be greatly influenced by the thermal gradient [22]. Several researchers have studied the probable impacts of a thermal bow in rotor systems, as well as methods for diagnosis of this defect.

To separate the vibrations resulting from an electrical failure and a bowed shaft, Bloch and Geitner [23] proposed an approach based on the vibration amplitude in time domain. Observing the motor immediately after power cutting off, a bowed rotor can be determined. Although vibration suspension immediately after disconnecting the power supply can be a sign of an electrical failure in the device, a gradual decrease in vibration can signify a bend. If the motor is tested quickly after starting, it is probable that no defect will be revealed. After operating temperatures and loads have been reached, temperature inspections of the equipment can be informative.

In diagnosing a thermal bow in a power unit generator, a model-based identification approach was used by Pennacchi and Vania [24]. During shutdowns and runups, the transient 1X vibrations were measured. A thermal bow was discovered using differences in 1X vibration amplitude during a sudden shutdown versus a planned shutdown. Bending moments were applied to the FE (finite element) model of the system to model a thermal bow on the rotor. Furthermore, by comparing experimental and modeled results, it was determined that the current fault is a bow rather than an unbalance. The accuracy of the fault identification was assessed by statistically analyzing the errors resulting from the experimental and theoretical responses.

Baldassare and Fontana [25] found that the thermal bend can be removed naturally if the rotating machine starts working. They also observed that the highest vibration level resulting from a bent shaft is measured when the highest temperature difference is measured.

To identify a thermal bow, Vania, Pennacchi, and Chatterton [2] dealt with a case study consisting of a combinatorial rotary machine made up of a low-pressure turbine, a high-pressure–intermediate-pressure steam turbine, and a generator. The flexural critical velocities of the first balance resonance were split in two. In the theoretical model, the faults were simulated by various equivalent forces and excitations. The difference between the vibrations measured during runup and rundown and in speeds close to the first balance resonance caused suspicion of a thermal root fault. On the 1X vibration, the effects of unbalancing and shaft bow appeared to be identical. A coherence factor that measured the error between experimental and theoretical vibrations was used to distinguish distributed unbalancing (shaft bow) from local unbalance; if the amplitude of the vibration changes with the temperature, the fault is assumed to be bow rather than unbalancing.

Peng et al. [26] evaluated the deflection of a bowed shaft caused by a thermal discrepancy in a rotary system in various conditions consisting of different temperatures. Several researchers have assayed the effects of an initial bend on the vibration amplitude of geared rotor systems, too.

Shiau et al. [27] studied the coupling effects of lateral and torsional motions in a bowed geared rotor-bearing system. The dynamic characteristics of the system were investigated, including natural frequencies, mode shapes, and steady-state response. The magnitude of the initial shaft bow, as well as its phase angle, had a significant impact on the system's natural frequencies and steady-state response, according to the findings. The residual shaft bow dramatically increased the system's steady-state response for the in-phase case when the spin speed was close to the second critical speed. Furthermore, when the system was set under special conditions, the system produced a zero response. The second critical speed of the system was affected by the phase angle of the residual shaft bow, too. When the phase angle was augmented from 0° to 90°, the second critical speed increased, and it decreased as the bow phase angle raised from 90° to 180°. The bow phase angle had a significant impact on the peak responses of the second mode, especially when the phase angle was increased from 90° to 180°. In the second mode, the effect of the shaft bow was greater than its influences in the third mode. The peak response for the higher modes was quite different due to the combined effect of the shaft bow and gear disc positions; the maximum peak response did not occur at zero phase angle due to the gyroscopic effect.

Kang et al. [28] investigated the natural frequencies, as well as the steady-state response of a geared rotor system equipped with viscoelastic bearings under the influence of gear eccentricity, gear mesh transmission error, and residual shaft bow. Six different scenarios were considered, based on the bow magnitude, phase angle, stiffness, and the amount of loss factor. The findings revealed that the features of the viscoelastic support, i.e., mass, stiffness, and loss factor, can vary the amplitude of critical speeds and steady-state response. The residual bow magnitude and its phase angle had a huge impact on the first critical speed. It was seen that as the residual bow increased, the peak value rose. When the magnitude of the residual bow was twice the disc eccentricity, the response remained constant regardless of spin speed. At a 45° phase angle, the maximum peak value was observed, but not at a 0° phase angle. When the spin speed was in the range of the valley between two resonance crests, the response was suppressed and smaller than the residual bow as the phase angle decreased. The counteraction of mass unbalance and bow effects neutralized the system response in this range; the resonance response was reduced due to the loss factor and stiffness of the viscoelastic support. The viscoelastic supports can reduce system vibration and extend the working range of the high-speed rotor. As the stiffness of the viscoelastic supports increased, the peak values of the lateral response witnessed a decrease in comparison to the one without viscoelastic supports; on the other hand, a rise in the stiffness of viscoelastic supports can improve the critical speed. When the system was equipped with viscoelastic supports, it was explored that one more mode was excited at low spin speed. They showed that as the mass of the viscoelastic support decreased, the critical speeds increased.

Chen and Kuo [29] studied how a spur-geared rotary system behaved dynamically. The system was thought to have translational motion due to the shaft deformation and initial bow. It was assumed that the distance between the central point of two gears is time-varying. Parallel with an increase in the initial shaft bow, the magnitude of the lateral response, contact ratio, and the gear pressure angle raised. If a higher external force will be applied to the system, the initial shaft bow can increase the gear pressure angle, and also the contact ratio. Conversely, if the phase angle of the initial shaft bow increases, the gear contact ratio diminishes.

Chen [30] investigated the translational movement of a two-stage geared rotor-bearing system under the effects of deformation in a spinning shaft. The contact ratio of the gear and pressure angle were considered time-dependent variables. The residual shaft bow increased the magnitudes of lateral vibrations, the phase angle of gear pairs, and the contact ratio in the geared system. This indicated that the presence of residual shaft bow had a significant impact on the system's lateral response, increasing amplitudes and thus destabilizing the system.

Multiple scholars have studied the impacts of a residual shaft in rotating machines equipped with fluid film bearings.

Salamone and Gunter [31] utilized the transfer matrix method for a multi-mass rotor system with fluid film supports suffering unbalance, shaft bow, and disc skew. The effects of these three faults in a water pump were illustrated. It was shown that because of the shaft bow and disc skew, the unbalance response at the critical speeds changed dramatically. The synchronous responses in the first three critical speeds were evaluated. It was claimed that the shaft bow was dominant (increased the amplitude) at the first critical speed, while the effects of both shaft bow and disc skew were visible at the second whirling speed. The effects of disc skew, on the other hand, were dominant at the third critical speed.

Theoretically and experimentally, Flack and Rooke [32] investigated the synchronous responses of an unbalanced and bowed rotor system. Effects of four different types of fluid film bearings, i.e., pressure dam, four-lobe, tilting pad, and two kinds of axial groove, on the response of the malfunctioned system were researched; they used the transfer matrix procedure to extract the equation of motion. When rigid supports are used and the shaft bow and unbalance have the same amplitude and a 180° phase difference, the response reaches zero at the critical speed. The vibration amplitude of fluid film bearings, on the other hand, is not exactly zero in the same situations but is nearly constant.

Meagher, Wu, and Lencioni [33] solved the equation of motion for a bowed rotor system consisting of fluid film bearing, then compared the responses in midplane and bearing points experimentally and theoretically. It was stated that there is no self-balancing speed in the case of fluid film bearing, but the system response is at its minimum magnitude. Although the peak amplitudes decreased when the disc was closer to the fluid bearing, moving the disc along the shaft length had no effect on the self-balancing speed. When measuring at the mid-plane, the low-speed shutdown was introduced as a good tool for recognizing the shaft bow, but measurements at the fluid bearing did not yield a reliable result. In contrast, phase measurements in mid-plane and fluid bearing at low speeds accurately represented bow angle.

The effects of a residual shaft bow on the onset of rubbing, as well as the contact force between the rotor and stator in rotating systems, have been examined too. Braut, Žigulić, and Butković [16] modeled the rotor–stator contact force in two ways, linearly and nonlinearly, and for various compositions of mass unbalancing and shaft bent. The resulting responses of the permanent contact were shown to be independent of the modeling assumption for the force in question, while for the first contact, the nonlinear contact hypothesis sowed greater response. The numerical results were tuned with the experimental ones. The supercritical and subcritical self-balancing speeds appeared when the amount of mass eccentricity was smaller and greater than the shaft bow amount, respectively. The contact between the rotor and the stator caused the rotor deceleration to be more intensive. Increasing the phase lag from 90° to 180°, the contact force experienced a rise.

A universal model of a rub-impact rotor-bearing system with a residual bow was expanded by Shen, Jia, and Zhao [34]. Due to an initial shaft bow in the rotor system, the speed at which rubbing began was reduced. When the system had a permanent initial bow, the synchronous full annular rub occurred. By increasing the response magnitude caused by this rub, chaotic motion occurred, and the system became unstable. In the absence of a residual shaft bow, the motion was period-doubling bifurcation, which resulted in chaos after the rubbing.

Sanches and Pederiva [35] suggested an extended algorithm based on a least-squares approximation to identify unbalancing and initial shaft bow in rotating machinery. The model-based methodology contained information about the amplitude and phase of the faulted systems. Without measuring the input excitations, the proposed methodology utilized the correlation analysis and the Lyapunov matrix equation in the identification procedure. To diminish the dimension of the state vector, the output measurement reduction was applied based on coordinate transformation in the Lyapunov matrix equation. In the reduction process, only linear measurements at the positions of discs were performed.

The same authors extracted the identification equation for an unbalanced and a bowed rotor system using the reduced system (performed by the Guyan reduction process) [36]. To investigate fault parameters, the least-square fitting method was used. The differential evolution optimization method was used to distinguish the bearing physical properties, as well as coupling and rotor damping. When the unbalance and shaft bow were in phase, distinguishing these two faults was difficult. Adding a noise, about 6% of the signal magnitude, affected the amplitude measurement, but phase measurement did not experience a significant change.

For malfunction diagnosing, dynamic trend analysis is a useful tool. A hierarchical presentation of signal trends, their extraction, and their comparison (estimation of similarity) to infer the state of the process are all part of trend analysis [37]. Moreover, performing smart fault diagnosis manners such as machine learning has increased sharply during the last few years.

Yang and Hsu [38] combined reasoning systems and trending analysis with machine learning to locate the shaft bow and unbalance, as well as the number of these faults in large rotor systems. Using a model-based approach that was earlier introduced in [39] and two statistical features, i.e., average and standard deviation, the location of local bows was identified.

Nembhard et al. [40] investigated discrepant damages in a rotor system, including rub, rotor bow, crack, looseness of bearing, misalignment at bearing, and added unbalance. The effects of each of these faults were assessed using orbit plots.

The effects of a residual bow and rub-impact in a rod fastening rotor system were investigated by Hu et al. [41]. Five different graphs—a bifurcation diagram, a time-domain waveform, an orbit plot, a frequency spectrum, and a Poincaré map—were drawn to evaluate the dynamic behavior. The analysis revealed that synchronous periodic-1 motion, multi-periodic motion, chaotic motion, and quasi-periodic motion were all present as a result of a rub-impact and a bent shaft. Additionally, the magnitude of the permanent bow has a discernible impact on the slow rotating speed, and as a result, as this value is raised, the system's rate of instability gradually rises.

Yang et al. [42] considered the influences of an initial bow and coupling damage of unbalance-rub in the rotor system. It was founded that the coexistence of the residual bow and the geometrical nonlinearity contributes to variations in the resonance characteristics. These changes were mainly affected by the initial bow's phase angle. Time-domain waveform, orbit diagrams, Poincaré map, frequency spectrum, and bifurcation diagrams were used to find the probable effects. Higher residual magnitude resulted in higher jumps in the whirling responses; effects of the residual bow were more patent at the low rotational speeds.

Smart Inspection

Using smart fault classification methods such as machine learning in rotor systems has increased sharply throughout the recent decade. Regression and classification are the two main methods that can be used to carry out this process. Furthermore, three techniques—supervised, unsupervised, and reinforcement—can be used to perform classification. Some features are necessary for all of these methods, regardless of whether users can manually enter these parameters or if the network can automatically extract the features. Take the principal component analysis (PCA) as an example [43].

Song et al. [44] studied the effects of the initial bow on the longitudinal responses of a rotary system. The system response was evaluated in three scenarios: with different amounts of residual bow magnitude, disc eccentricity, and damping ratio. To investigate the effect of the fault more accurately, one of the shafts was placed motionless in the test rig for two years. Wavelet transformation and nonlinear manifold learning processes were used in extracting the fault features. ISOMAP, a nonlinear dimensionality reduction procedure, along with PCA, were employed to reduce the dimension of the collected signal, and also to isolate signals of unbalancing and shaft bow. The eccentricity effect was found to inhibit

rotor vibration. When the eccentricity and damping ratio were small, the residual shaft bow could have a significant impact on rotor vibration. On the other hand, the impact was minimal. It was emphasized that the operating frequency is effective too. Prior to the resonance, the rotor longitudinal responses increased as the damping ratio decreased. The longitudinal responses of the rotors significantly decreased after the resonance.

Singh and Kumar [45] extracted six statistical features of time-domain signals that were investigated to classify unbalanced, misaligned, and bowed systems in the cases of single and/or multiple defects. An artificial neural network (ANN) and support vector machine (SVM) were used to process the extracted features. The results showed that the SVM took less time and effort to calculate.

Dhini et al. [46] classified four different faults, i.e., misalignment, bow, blade erosion, and cracked case, in a steam turbine of a thermal power plant. A multilayer perceptron neural network was applied in three various scenarios, including feature extraction and dimension reduction utilizing PCA.

Liu et al. [47] used a residual convolutional neural network for the classification of malfunctioned rotary systems. Six health scenarios, i.e., normal, bowed, broken bar, faulty bearing, high impedance, and unbalanced rotor systems, were considered. The nonstationary parts of the vibration signals were segmented, then for each section, the act of feature extraction was performed by the network automatically.

Considering the referenced articles, it can be understood that a wide spectrum of them investigated the impacts of a bowed shaft on the dynamic behavior of the complicated systems (with different supports or geared ones) rather than the simple Jeffcott rotor. Furthermore, the effects of a bowed shaft on the symptoms of other faults, such as disc unbalance, disc skew, and rubbing, have been discussed in some articles.

2.1.2. Frequency Domain

The most common tool in industrial applications for condition monitoring of rotary machines is the Fourier transform (FT). This method provides indications of the amount of a distinct frequency band in a signal [17]. It should be emphasized that methods such as Bode plot, Nyquist plot, and order analysis, which are based on the frequency response or tracking the higher frequency components, are placed in this category.

Experimental observations in several comprehensive books and handbooks expressed that the incidence of higher-frequency harmonics (further than the 1X) is a sign of a bowed rotor system.

In [48], Mobley stated that according to the graphs plotted using the fast Fourier transform (FFT), the 1X frequency component is the typical symptom of a bowed, and also a misaligned, rotor system. The 2X component, as well as the 3X and 4X components, may appear frequently depending on the severity of the fault and its location. Radial and axial vibrations are appropriate for the detection of these two faults.

Beebe [49] claimed that the 1X frequency component is visible in the frequency spectrum due to faults such as unbalance, bowed shaft, misalignment, bearing clearance, bearing looseness, and so on. Furthermore, the second frequency harmonics, 2X, can be produced by misalignment, loose internal components, and sometimes a thermally bowed shaft.

Girdhar [50] declared that, in addition to analyzing vibration signals in the time domain, where both the radial and axial amplitudes increase and a 180° phase difference exists in these two directions, the frequency domain should be considered as symptoms of a bowed rotor system. Because of a shaft bend in this domain, 1X and 2X frequency harmonics can be seen; the magnitude of the 2X component can lead us to the approximate location of the bow.

Furthermore, some authors evaluated the effects of a bowed shaft in the frequency domain theoretically and experimentally. Leader, Flack, and Allaire [51] investigated the stability and unbalance response of a bowed rotor system with three different types of supports. A bow in the shaft was revealed by the presence of 2X and 3X frequency

harmonics. Other small peaks appeared at various speeds although they were ascribed to resonances (structural ones) from the driving motor and noise, as well as coupling. Oil whip occurred at approximately twice the first rotor critical speed. Axial groove bearings were better for low-speed use and near the critical speed. At the first critical speed, the pressure dam bearing exhibited controlled vibration, but a whip occurred at twice and three times the critical speed. Tilting pad bearings, on the other hand, showed increased vibration near the critical speed but did not whip at the greater speeds.

Flack et al. [52] compared the unbalance response in the presence of mechanical/electrical runout, and shaft bow for a Jeffcott rotor model experimentally and theoretically. A compensation vector was extracted from the response at a low speed where the dynamic effects were negligible. The low-speed compensation was subtracted from the responses in other speeds to distinguish bowed shaft response. To study the responses, Bode (for the synchronous responses) and Nyquist (for the complex responses) plots were created. The maximum magnitude of vibration for a bowed rotor was dramatically dependent on the bow phase angle (angle between the bow and mass unbalance), whereas the response of the rotor with a runout had a poor dependency on the angle between the rotor runout vector and timing mark. In contrast with the bowed rotor, the minimum response in the case of the runout rotor was not zero. In the Nyquist plot, the shape of the response for the bowed rotor was close to a circle. In the case of the bowed rotor, the size of the response circles was a function of the bow phase angle, whereas the size of the runout system was not related to its phase angle. Theoretically, an instantaneous phase change (180°) should occur when the bow phase angle is 180°, whereas in the experimental response, this did not occur. It was claimed that for an unbowed rotor with no runout, the system damping can be determined by measuring the slope of the phase angle in a Bode plot at the critical speed. Without compensation, such a procedure for a bowed rotor frequently results in severe inaccuracies. However, it is projected that adopting such compensation will reduce mistakes to 5% or less.

To identify multiplex defects, including the shaft bow, unbalancing, and coupling misalignment in a rotating machine, a model-based procedure was developed by Bachschmid, Pennacchi, and Vania [39]. They employed a least-squares spectral analysis in identifying miscellaneous defects in the rotor system. The system was modeled by means of the model-based approach. For this purpose, the residual multidimensional vibrations (residual map) between measuring planes and the calculated vibrations resulting from the acting malfunctions were minimized. This procedure could identify the position, module, and phase of the damages.

Darpe, Gupta, and Chawla [53] modeled the equation of motion of a cracked rotor system that suffered a bowed rotor, too. They analyzed the steady-state and transient responses of the defected system. The goal of this research was to observe how the residual bow affects the stiffness of a rotating cracked shaft and what changes, if any, a bow could make to the dynamics of a cracked rotor. The stiffness variation was not significantly influenced by the usual level of the bow, and the nonlinear nature of the crack response was not significantly altered. When the phase angle between bow and crack was zero, the bow tended to keep the crack open. The bow tended to close the crack when the bow phase was 180° opposite the crack direction. However, the above results showed that if a bow existed out of phase with the crack, the presence of a crack might not cause any changes in the orbit of the asymmetric rotor. The bow, on the other hand, completely obscured the sensitivity of the orbital response of a cracked rotor to unbalance phase at half the critical speed, and the influence of the unbalance phase on the orbital response of a cracked rotor at half the critical speed could not be used for detection. Higher harmonics' directional nature and amplitude, such as the 3X frequency component, did not change significantly. The zero response at the self-balancing speed of the bowed rotor for a specific phase of unbalance was not observed due to the presence of a crack and the influence of gravity.

Khaire [5], in an experimental work, discovered that 1X and 2X vibration components appeared as a result of a bow in the shaft of a rotating system; a high level of the first axial vibration, i.e., 1X was observed as a sign of the fault in the time domain. Because these

harmonic components, i.e., 1X and 2X, were visible at various rotating speeds, the presence of these signs was interpreted to be independent of the system's speed, but by increasing the amount of the system's speed, the vibration magnitude experienced a rise.

A vibration analysis technique based on order analysis was proposed for bent shaft diagnosis by Mogal and Lalwani [54]. Both phase and amplitude were obtained through order analysis. The fault type (whether unbalancing or shaft bow) and location were determined based on phase and amplitude. The amplitude of the bent shaft's order spectra was higher at the first order, 1X. The bent shaft was confirmed by the phase difference, which was approximately 180° between the axial directions at the driver and driven end bearings.

Sarmah and Tiwari [55] evaluated a cracked–bowed–unbalanced rotating machine that was equipped with an auxiliary active magnetic bearing in a model-based manner (using the least-square fitting procedure). To achieve this, the full-spectrum harmonics, Nyquist, and Bode diagrams were considered. Based on the discovery, an initial bow in a cracked rotor system can disguise the symptoms of the crack by influencing the local flexural stiffness of the element containing the damage.

Wang, Xiong, and Hu [56] investigated the effects of an initial bow on the breathing behavior of a cracked rotary system. Increasing the magnitude of the residual bow, the range of the resonance area and also its amplitude raised. Furthermore, the initial bow disappeared with the frequency components that were created due to the crack.

Time-domain, frequency-domain, and orbit plot analyses were performed for an internally damped rotor system to identify multi-faults, i.e., unbalance, crack, and initial bow, by Sarmah and Tiwari [57]. The rotary system was equipped with an auxiliary active magnetic bearing. Both theoretical modeling and experimental validation were performed in the work. The initial shaft was identified using the additive stiffness of the crack. In the approach, the severity of the residual shaft bow changed the additive stiffness.

### 2.1.3. Time–Frequency Domain

Using the frequency analysis also has its cons, too. Since time information is not available in this domain, it is impossible to determine the precise time point or time period at which a frequency component changed. The use of time–frequency-domain analysis has been increased to address this [58]. In both manual and automatic fault diagnosis, time–frequency-domain analysis, such as wavelet transformation, has become a powerful tool. This method is particularly effective at distinguishing nonlinear features, for instance, in revealing crack symptoms in rotor systems [59,60].

Several more researchers have also used the time–frequency-domain transformations to diagnose bowed rotor systems. In [61], Srinivas, Srinivasan, and Umesh compared the effectiveness of the frequency-domain and time–frequency-domain (wavelet) transformations in the classification of unbalanced, bowed, and unbalanced–bowed rotor systems by means of the ANN. In the case of the frequency domain, amplitudes of frequency components of RMS (root mean square) of vibration velocity (mm/s) were selected as the input data to the ANN. On the other hand, the RMS of the detail coefficients of wavelet transformation were employed as the input data. In both cases, the achieved accuracy was above 99%, i.e., 99.81% and 99.9% in the cases of frequency and time–frequency domain, respectively.

A combined approach consisting of continuous wavelet transform (CWT) and rough set theory was employed by Konar et al. [62] in the classification of seven rotor systems, including a healthy induction motor, also with faulted bearings, stator fault, and voltage unbalance, as well as with a faulted shaft, i.e., bowed, broken, and unbalanced. The CWT spectrograms were used to extract statistical features such as mean, RMS, variance, standard deviation, crest, kurtosis, and entropy. Furthermore, the rough set procedure was used to select the most effective scales during the feature selection step, reducing the required computing time by 85%. The classification process was carried out using three supervised classifiers: multilayer perceptron (MLP), radial basis function (RBF), and support vector machine (SVM).

Singh and Kumar [63] decomposed the complex vibration signals of a bowed and unbalanced rotor system into a number of intrinsic mode functions using the empirical mode decomposition technique. The Hilbert spectrum was used to reveal the characteristics of these two damages following decomposition. The initial bow's impacts changed depending on its phase angle. Higher vibration amplitude and wider intra-wave frequency modulation band were observed when the phase angle of the residual bow was 0°. In contrast, when the bow's phase angle was equal to 180°, the vibration amplitude and frequency band were both diminished.

Grezmak et al. [64] used a deep convolutional neural network (DCNN) to classify rotor systems into four categories: healthy, bowed shaft, broken shaft, and unbalanced. Spectrograms of continuous wavelet transformations with Morlet as the mother function were introduced as input data. In addition, a layer-wise relevance propagation (LRP) post-training procedure was used to interpret the classification results.

### 2.2. Balancing/Correcting

While the existence of some faults, such as a crack in the rotating system, means the defected component should be replaced, some other defects, such as unbalance, misalignment, and bowed shaft, if diagnosed early, can be resolved.

Although both a bent shaft and an imbalance produce synchronous whirls, these movements and the consequent balancing procedures are different. Differences in the force effects created by these faults could be the cause of these discrepancies. The effect of the force produced by an initial bend does not change with shaft speed and is a function of the shaft stiffness matrix and the initial bow vector, while the effect of the force produced by an imbalance is proportional to the square of the angular speed of the shaft [65].

For balancing a bowed rotor system, two main procedures have been used by scholars: influence coefficient and modal balancing method. In the following, the related investigations are referenced.

### 2.2.1. Influence Coefficient Method

As a primary work in balancing bowed rotor systems, Nicholas, Gunter, and Allaire [66] employed three procedures based on the influence coefficient method. At the balance speed, the first method brought the whole shaft amplitude to zero. The second approach balanced the elastic deflection to zero, leaving the residual bow amplitude. Without rotating the rotor at the critical speed, the third way balanced the total shaft amplitude to zero. Nelson [67] expanded the method mentioned in [54] for a multi-degree-of-freedom rotor system. Furthermore, influences of unbalancing and shaft bow on the steady-state response of the system were evaluated. Deepthikumar, Sekhar, and Srikanthan [68] extracted the required influence coefficients for balancing a bowed rotor system by means of the transfer matrix.

### 2.2.2. Modal Balancing Method

On the other hand, the modal balancing approach has attracted more attention in balancing bowed rotor systems. Gnielka [69] applied the modal balancing approach for a multi-bearing rotary system (consisting of fluid-film bearing). It was claimed that bowed systems with ball bearings, and also with fluid film bearings, can be balanced with this procedure. Impacts of the number of measuring points, and also their location on the balancing procedure, were investigated.

Parkinson, Darlow, and Smalley [65] balanced a warped rotary machine using the modal balancing method analytically and experimentally. Balancing a bowed rotor was performed according to the net whirl instead of the total whirl and beyond the fourth flexural (critical) speed.

To balance a bowed rotor system, Meacham et al. [70] presented a complex modal balancing procedure. Although a mathematical model is required to obtain the system's modal parameters, the method's main advantage was claimed to be the lack of the need for

trial runs. A working steam turbine generator and a gas turbine system with computer-generated response data were both used to test the proposed process. Three scenarios were assumed for the balancing process. First, the initial bow and single mass unbalancing behaved as a tantamount imbalance. Second, the system dynamic response at the balance velocity was subtracted from the low-level pace in the runout. Finally, the balancing procedure was completed at the critical speed by using the neighboring speed.

Deepthikumar, Sekhar, and Srikanthan [71] used the model discussed in [72] for the distributed unbalance to balance a bowed rotor system at its first bending critical speed. They employed the modal balancing procedure, a single trial run, and a single balancing plan. Knowing the amplification factor at a critical speed, the modal correction mass required to balance a rotor at its first bending critical speed was calculated.

### 2.2.3. Optimization Method in Correcting

Besides the aforenamed balancing approaches, optimization procedures can be used to diminish a bowed rotor vibration level. Kang [73] modified the effects of an initial shaft bow in a geared rotor/bearing system by considering the critical speed and using a genetic algorithm and an augmented Lagrange multiplier as part of an optimization algorithm. The best position for bearing supports to reduce residual bow effects was determined using this method.

Chen, Jiang, and Liu [74] proposed an optimal balancing procedure for the steam turbine rotor. At first, the cause of the bow was determined as uneven creep influence. Then, using an optimized technique, the whirling mode of the system was affected. In employing this process, not only was the amplitude of vibration deceased noticeably, but it also diminished the advancement of the bow.

## 3. Results

This paper aims to review almost all the literature in the field of bowed rotating system diagnosis and balancing approaches.

It can be seen from the cited articles that certain symptoms have been identified more frequently than others in the field of research on the dynamic behaviors of bowed rotor systems, including the diagnosis of this damage. Increased vibration amplitude (commonly in comparison to an unbalanced system) is found to be the most common sign of a bowed rotor system in the time-domain analysis, while zero-vibration amplitude, sometimes referred to as self-balancing speed, is found to be the second most prevalent feature. The last one can happen in a particular circumstance when the magnitudes of the shaft bow and disc eccentricity are similar, and these two faults also have a 180° phase difference. It should be emphasized that in some works, more than a single feature was detected.

On the other hand, in the frequency domain, the most prevalent indicator of a bowed rotor system is the presence of the 1X and 2X harmonic components in the frequency graph. Moreover, the existence of the higher component, i.e., 3X, was discovered as a sign of a bowed system too.

The referenced articles in which the most common signs of bowed rotating systems were discovered both in the time and frequency domain are represented by Figure 4 and its companion table, Table 1. In the aforementioned table, works are arranged chronologically, and some references are mentioned more than one time since in them, two or more domains were investigated. It is important to remember that until now, no distinct signature was identified in the time-frequency domain, and only statistical features were extrapolated from the transformed signals to this domain.

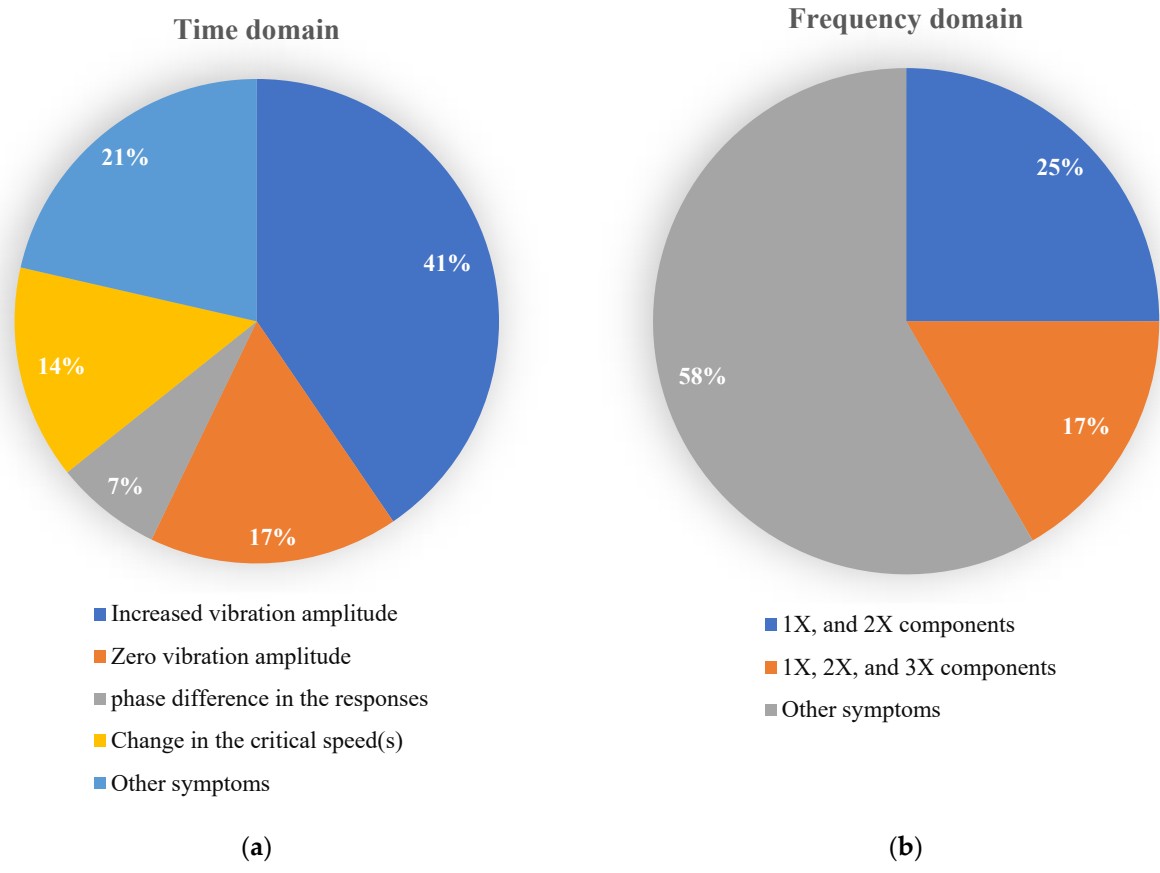

**Figure 4.** Most common symptoms of bowed rotor systems: (**a**) time domain, (**b**) frequency domain.

**Table 1.** Most common symptoms of bowed rotor systems.

| Researcher(s) | Investigated Fault(s) | Type(s) of Research | Symptom | Domain |
|---|---|---|---|---|
| Nicholas, Gunter, and Allaire [6] | Unbalancing and 'shaft bow | Theoretical | | |
| Shiau and Lee [4] | Unbalancing, shaft bow, and disc skew | Theoretical | | |
| Flack and Rooke [32] | Unbalancing and shaft bow | Theoretical–experimental | | |
| Shiau et al. [27] | Shaft bow (geared system) | Theoretical | Zero-vibration amplitude (self-balancing speed) | Time |
| Kang et al. [28] | Unbalancing and shaft bow (geared system with viscoelastic supports) | Theoretical | | |
| Braut, Žigulić, and Butković [16] | Unbalancing, shaft bow, and rubbing | Theoretical–experimental | | |
| Flack et al. [52] | Unbalancing, shaft bow, and mechanical/electrical runout | Theoretical–experimental | | |
| Edwards, Lees, and Friswell [19] | Unbalancing, shaft bow | Experimental | | |
| Girdhar [50] | Shaft bow (based on the industrial cases) | Experimental | Phase difference in the responses | |
| Mogal and Lalwani [54] | Unbalancing, shaft bow | Experimental | | |

**Table 1.** *Cont.*

| Researcher(s) | Investigated Fault(s) | Type(s) of Research | Symptom | Domain |
|---|---|---|---|---|
| Shiau and Lee [4] | Unbalancing, shaft bow, and disc skew | Theoretical | | |
| Rao [20] | Unbalancing, shaft bow | Theoretical | | |
| Lin and Lei [3] | Shaft bow | Theoretical | | |
| Pennacchi and Vania [24] | Unbalancing, thermal shaft bow | Theoretical–experimental | | |
| Baldassare and Fontana [25] | Thermal shaft bow | Theoretical–experimental | | |
| Vania, Pennacchi, and Chatterton [2] | Thermal shaft bow | Theoretical–experimental | | |
| Shiau et al. [27] | Shaft bow (geared system) | Theoretical | | |
| Kang et al. [28] | Unbalancing and shaft bow (geared system with viscoelastic supports) | Theoretical | Increased vibration amplitude | Time |
| Girdhar [50] | Shaft bow (based on the industrial cases) | Experimental | | |
| Chen and Kuo [29] | Shaft bow (spur-geared system) | Theoretical | | |
| Chen [30] | Shaft bow (two-stage geared system) | Theoretical | | |
| Meagher, Wu, and Lencioni [33] | Shaft bow (with fluid film supports) | Theoretical–experimental | | |
| Shen, Jia, and Zhao [34] | Shaft bow and rubbing | Theoretical | | |
| Flack et al. [52] | Unbalancing, shaft bow, and mechanical/electrical runout | Theoretical–experimental | | |
| Khaire [5] | Shaft bow | Experimental | | |
| Singh, and Kumar [63] | Unbalancing, shaft bow | Theoretical–experimental | | |
| Salamone and Gunter [31] | Unbalancing, shaft bow, and disc skew | Theoretical | | |
| Shiau and Lee [4] | Unbalancing, shaft bow, and disc skew | Theoretical | Change in the critical speed(s) | |
| Rao [20] | Unbalancing, shaft bow | Theoretical | | |
| Kang et al. [28] | Unbalancing and shaft bow (geared system with viscoelastic supports) | Theoretical | | |
| Yang et al. [42] | Shaft bow, unbalancing-rub | Theoretical | | |
| Beebe [49] | Shaft bow (based on the industrial cases) | Experimental | 1X and 2X harmonic components | Frequency |
| Girdhar [50] | | Experimental | | |
| Khaire [5] | Shaft bow | Experimental | | |
| Leader, Flack, and Allaire [51] | Shaft bow, unbalancing | Experimental | 1X, 2X, and 3X harmonic components | |
| Darpe, Gupta, and Chawla [53] | Shaft bow, shaft crack, unbalancing | Theoretical | | |

## 4. Discussion

Perusing the prior sections, some cases can be raised in the practical study of the diagnosis and balance of bowed rotating devices:

- The time-domain method, which has primarily been used in troubleshooting bowed rotors, is not widely applicable in the diagnosis of some other faults, such as cracks. As a result, forming a feature vector would be problematic if an intelligent method, such as neural networks, is desired to isolate several faults.

- In different investigations, the introduced symptoms of a bowed rotor system in the FFT plot were somewhat different. Increased 1X frequency components were stated to be a sign of a bow in some works, while 2X, 3X, and even 4X components were

mentioned in others. It appears that the effects of various physical components, such as coupling (geared or flexible), on revealing higher components should be considered.

- Although the influences of a shaft bow on the behavior of some other faults, such as crack, rub, misalignment, and disc skew, were studied, the effects of this defect on other malfunctions, such as ball-bearing faults or mechanical losses, should be investigated further.
- Some straightening procedures are used in the industry, such as determining the location of the bow peak and straightening the rotor with a press machine or, in some cases, thermal methods, while they have not been studied in the academic context.
- While time–frequency-domain techniques, such as wavelet transformation, have become effective tools for diagnosing a variety of damages in rotating systems, the signs of a bent shaft in this domain have not yet received sufficient attention. Therefore, a thorough wavelet analysis of bowed rotor systems using CWT or discrete wavelet transformation (DWT) is worthwhile as prospective research.

The diagnosis of bent shafts in rotor systems has largely relied on vibration analysis, but other methods, such as thermography, can also be used. Additionally, analyzing current signals can be used to study bowed induction motors, which is outside the scope of this study.

## 5. Conclusions

Although some preventative measures can be taken to avoid a rotor system becoming bowed, such as working in turning gears, a primitive diagnosis of a bowed rotor system, as well as appropriate balancing/correcting approaches, are critical in reducing the occurrence of calamitous failures and serious economic fatalities. This article provides an overview of bowed rotary machine diagnosis and balancing/correction procedures. The work in the field of diagnosis is classified in this review based on the domain in which the analysis was performed, i.e., time, frequency, and time–frequency domain. The influence coefficient, modal balancing, and the optimization method in correcting are all considered in the scope of the balancing/correcting section. Finally, some current doubts and also future research possibilities are listed. This review paper is intended to serve as a road map for those interested in researching the quality of the diagnosis and balancing of bowed rotating systems.

According to the findings, an increased vibration amplitude can be introduced as the major feature of a bowed rotor system in the time domain, although in the frequency domain, higher-order frequency components can be informative.

Unlike the balancing/correcting section, where the number of previous works is quite limited, in the case of the diagnosis of the bowed rotor systems, many studies have been performed. A wide range of them were carried out in the time domain, where the analysis of some other faults, such as cracked shafts that show a nonlinear behavior, is relatively inconclusive. Therefore, focusing on time–frequency techniques can be promising in future research that is supposed to distinguish multiple malfunctions.

The modal balancing method has the grandest market share in the balancing area; optimization techniques, however, appear to have recently become more popular.

**Author Contributions:** Conceptualization, N.R.; research, N.R., A.D.L., G.L. and F.C.; data curation, N.R., A.D.L., G.L. and F.C.; writing—original draft preparation and review and editing, N.R., A.D.L., G.L. and F.C.; supervision, F.C. All authors have read and agreed to the published version of the manuscript.

**Funding:** This research received no external funding.

**Institutional Review Board Statement:** Not applicable.

**Informed Consent Statement:** Not applicable.

**Data Availability Statement:** Not applicable.

**Conflicts of Interest:** The authors declare no conflicts of interest.

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
