# Peer review of "Diagnosing and Balancing Approaches of Bowed Rotating Systems: A Review"

_applsci, doi:10.3390/app12189157_

Round 1

Reviewer 1 Report

This paper represents a review of almost all the investigations and studies that have been done on the diagnosing and balancing of bowed rotating systems, which include the diagnosing and balancing/correcting approaches to bowed rotors, and a summary is provided to the new research prospects.

I recommend this work for publishing in this journal after making a minor revision according to the following comments.

1) The researches need to be further subdivided and classified

2) More charts and formulas are needed to show the research results.

Author Response

Dear reviewer,

In the following, the method of applying the requested amendments is described.

Regards,

  • Two extra sub-division are added to the section of "Time domain" called Manual inspection, and Smart inspection with section numbers 2.1.1.1, and 2.1.1.2, respectively;
  • The explanation of the equation of motion in the case of a bowed rotary system, as well as the companion symbols, may be beyond the scope of this article. As a result, no formula has been added to the article. However, the meanings of the symbols in figure 1 have been clarified, and a relevant reference for the formulation of such a system has been addressed, namely reference [6]. It should be noted that Table 1 has been updated by enhancing the number of columns. An additional column reveals the type of the research, e.g., Theoretical, Experimental, and both.

Reviewer 2 Report

This paper presents a review of investigations and studies that have been done on the diagnosing and balancing of bowed rotating systems. However, the investigation of the paper is not comprehensive, and there is a lack of references in recent years. For example, the citations of the following two papers are 219 and 142 respectively (from Google academic), but this paper does not mention these two works.

[1] Gao, Q. , et al. "Rotating machine fault diagnosis using empirical mode decomposition." Mechanical Systems & Signal Processing 22.5(2008):1072-1081.

 [2] Li, X. , et al. "Diagnosing Rotating Machines With Weakly Supervised Data Using Deep Transfer Learning." IEEE Transactions on Industrial Informatics 16.3(2020):1688-1697.

This is just an example. Of course, there are other important research work to be supplemented.

1.     The work of bowed rotor diagnosis is divided into three major categories in this study: time-domain, frequency-domain, and time-frequency-domain. These three categories are for vibration signals. In addition to the vibration-based method, are there other methods that can be used for bowed rotating system diagnostics?

2.     Since many research works are listed in the paper, there is a lack of a more detailed technical roadmap for readers to browse.

3.     Please explain the symbols in Figure 1. What is the purpose of the rectangular box on the left side of Figure 1?

4.     What is the order of the research work listed in Table 1? Why are references [4] [41] [25] repeated in Table 1.

5.     No superscript is required when there is only one affiliation of the authors.

Other issues in the paper will not be listed one by one. This paper needs to be completely revised.

Author Response

Dear reviewer,

In the following, the method of applying the requested amendments is described.

Regards,

The two mentioned articles have been referenced; a number of other research papers have been accompanied too.

  • It has been clarified that these three techniques can be used in all signal processing procedures not only in vibration-based signals. However, it has been mentioned that other methods like thermography can be used. Take current signal analysis in induction motors as a striking example;
  • As a remedy for this issue, two sub-sections have been added; an extra column explains the type of the previous research.
  • Not only the vague symbols in Figure 1 have been explained, but also the left-side graph in the same figure has been modified by stating the rectangular box is the central disk;
  • It has been stated that the aforementioned references had repeated because in them more than one symptom had been found, or they had been performed in both time and frequency domains;
  • The mistake in addressing the affiliation has been corrected.

Reviewer 3 Report

Not enough comparative studies were included in the Balancing and correcting section, more research could be added.

Studies have been discussed in detail in the diognasis section, but comparative explanations should also be included.

The same references are repeated in Table 1. In addition, the columns in Table 1 should be arranged to give more information.

The conclusion part should be rearranged in a way that includes the results more clearly.

Author Response

Dear reviewer,

In the following, the method of applying the requested amendments is described.

Regards,

  • Since the number of previous investigations concerning the balancing of bowed rotary systems is limited, it is not possible for adding new references in this section. Also, one more article has been added to the part. If the respected reviewer knows a suitable paper, the authors would be pleased to add it to the article;
  • Three extra paragraphs have been subjoined to the conclusion part for the comparative discussion;
  • One extra column has been added for giving further information in Table 1; It has been stated that the aforementioned references had repeated because in them more than one symptom had been found, or they had been performed in both time and frequency domains;
  • Some more explanations have been added in the conclusion section. 

Round 2

Reviewer 2 Report

Please explain in detail what has been modified in the revised manuscript. As stated, "a number of other research papers have been accompanied", What are the research papers? "two sub-sections have been added", Which two sub-sections?" The same as other responses.

Please respond to the review comments one by one in detail. 

Author Response

Regarding the paper entitled " Diagnosing and balancing approaches of bowed rotating systems: A review", for publication in Applied Science journal and the following explanations are sent for further clarification.

  • 13 extra research have been referenced, i.e., [10-12], [40-43], [46-47], [55-56], [63], and [74].
  • It has been clarified that these three techniques can be used in all signal processing procedures not only in vibration-based signals. However, it has been mentioned that other methods like thermography can be used. Take current signal analysis in induction motors as a striking example.
  • Two extra sub-division are added to the section of "Time domain" called Manual inspection, and Smart inspection with section numbers 2.1.1.1, and 2.1.1.2, respectively. Moreover, the flowchart, Figure 2 has been updated in this regard; an extra column explains the type of the previous research.
  • Not only the vague symbols in Figure 1 have been explained, but also the left-side graph in the same figure has been modified by stating the rectangular box is the central disk.
  • It has been stated that the aforementioned references had repeated because in them more than one symptom had been found, or they had been performed in both time and frequency domains.
  • The mistake in addressing the affiliation has been corrected.

Reviewer 3 Report

Corrections have been made.

Author Response

Dear reviewer,

In the following, the method of applying the requested amendments is described.

Regards,

  • Since the number of previous investigations concerning the balancing of bowed rotary systems is limited, it is not possible for adding new references in this section. Also, one more article, [74], has been added to the part. If the respected reviewer knows a suitable paper, the authors would be pleased to add it to the article;
  • Three extra paragraphs have been subjoined to the conclusion part for the comparative discussion;
  • One extra column has been added for giving further information in Table 1; It has been stated that the aforementioned references had repeated because in them more than one symptom had been found, or they had been performed in both time and frequency domains;
  • Some more explanations have been added in the conclusion section;
  • 13 extra research have been referenced, i.e., [10-12], [40-43], [46-47], [55-56], [63], and [74].
  •